# Characterization of Expression and Epigenetic Features of Core Genes in Common Wheat

**DOI:** 10.3390/genes13071112

**Published:** 2022-06-21

**Authors:** Dongyang Zheng, Wenli Zhang

**Affiliations:** State Key Laboratory for Crop Genetics and Germplasm Enhancement, Collaborative Innovation Center for Modern Crop Production Co-Sponsored by Province and Ministry (CIC-MCP), Nanjing Agricultural University, No.1 Weigang, Nanjing 210095, China; 2018201004@njau.edu.cn

**Keywords:** core genes, expression, epigenetic features, *Triticum aestivum*

## Abstract

The availability of multiple wheat genome sequences enables us to identify core genes and characterize their genetic and epigenetic features, thereby advancing our understanding of their biological implications within individual plant species. It is, however, largely understudied in wheat. To this end, we reanalyzed genome sequences from 16 different wheat varieties and identified 62,299 core genes. We found that core and non-core genes have different roles in subgenome differentiation. Meanwhile, according to their expression profiles, these core genes can be classified into genes related to tissue development and stress responses, including 3376 genes highly expressed in both spikelets and at high temperatures. After associating with six histone marks and open chromatin, we found that these core genes can be divided into eight sub-clusters with distinct epigenomic features. Furthermore, we found that ca. 51% of the expressed transcription factors (TFs) were marked with both H3K27me3 and H3K4me3, indicative of the bivalency feature, which can be involved in tissue development through the TF-centered regulatory network. Thus, our study provides a valuable resource for the functional characterization of core genes in stress responses and tissue development in wheat.

## 1. Introduction

The advent of pan-genomes helps to reveal genomic-wide variations and conservation of genomic DNA sequences within different varieties of a species, which cannot be fully achieved by analyzing a single reference genome [1,2]. The first pan-genome of eight different strains of *Streptococcus agalactiae*, a pathogenic species isolated from humans, was initiated by Tettelin and colleagues [3]. The concept of pan-genomes was expanded to animal species [4]. It has recently been surged in a plethora of plant species due to their rich resources of distinct germplasms and low high throughput sequencing cost [4,5], including barley, *Brachypodium distachyon*, *Brassica oleracea (B. oleracea)*, *Brassica napus*(*B. napu*), rice, soybean, cotton, maize, strawberry, tomato, and wheat [6,7,8,9,10,11,12,13,14,15]. Comprehensive characterization of pan-genomes can greatly advance genomic understanding of a plant species and related evolution studies, and help to mine key genomic loci as key bioengineering targets for crop improvement [16,17]. For example, pan-genomes were widely applied for the identification of structural variations (SVs) associated with functional genomic loci responsible for certain agronomic traits, including the regulatory elements of the *Miniature seed1* gene in maize [8], the seed luster gene in soybean [9] and the leaf-sensitive gene in rice [6]. According to genomic sequence conservation and diversity across the population, pan-genomes can be divided into core genomes (conserved regions) and dispensable genomes (diverged regions) [18]. Conceptually, core genomes represent genes and other genomic loci that are common in the vast majority of plant and animal populations, while dispensable genomes indicate that genes and other genomic loci are highly variable across plant and animal populations. The number of core genes was found to be gradually decreased while the number of pan-genes was gradually increased as increase of the genome of a species [17]. It has been reported that core genes have distinct structures, evolutionary rates, and gene selection pressures (Ka/Ks) as compared to dispensable genes. Briefly, core genes are longer [10], more functionally conserved [17] and evolutionarily faster [19]; they have less DNA polymorphisms [11,20,21] and lower selection pressures [11,12], higher gene expression levels [11], and lower tissue specificity [8]. More importantly, core genes were found to be involved in some critical biological processes [11,17], indicative of their importance across population.

Wheat is the largest food crop grown in the world, providing approximately 47% of food calories in the world [22]. It contains three subgenomes (A, B and D) with total genome size of about 16 Gb and 80% of which are highly repetitive DNA sequences [23]. Due to a high genome complexity, extensive pan-genome studies lag far behind in common wheat as compared to other crops like rice, maize, and soybean. The first pan-genomes of 18 wheat varieties completed in 2017 revealed that there were 128,656 total pan-genome related genes and approximately 81,000 core genes [24]. A subsequent wheat pan-genome of resequencing 145 wheat varieties identified 15,552 new genes [25]. With recent completion of the wheat 10+ genome sequencing, the rich pan-genomes of common wheat pave the way to boost genomic studies across the wheat population [26,27]. For example, a haplotype-driven approach is proposed to dissect the 10+ genome to improve the precision of wheat breeding [28]. However, the role of subgenome differentiation, biological implications of core genes, and their epigenetic features in common wheat have few studies. To address this, we specifically analyzed pan-genomes of 16 wheat varieties, then identified core genes for functional characterization and association studies with histone marks.

## 2. Materials and Methods

### 2.1. Identification of the Core Genome of Common Wheat

We collected 16 wheat genomic protein files from the *ensemble plants*. https://plants.ensembl.org/Triticum_aestivum/Info/Index (accessed on 1 May 2022). The core genes were then identified using the core cruncher with the default parameters [29]. The detailed parameters are python corecruncher_master.py -in input_folder -out output_folder -length 80% -score 90%. Briefly, we used the protein sequences of Chinese Spring as a reference. The protein sequences were aligned using usearch & mafft [30,31]. The protein sequences with more than 90% similarity, 80% coverage of the protein length, and present in at least 80% of the wheat varieties were considered as core genes and retained for further analyses. The gene ids of these genes were extracted for the Chinese Spring.

### 2.2. Gene Expression Patterns and Epigenetic Landscapes of Core Genes

To examine the expression profile of core genes, we downloaded public RNA-seq data generated from different tissues and different treatments [32]. We excluded genes with a sum of FPKM less than 3 for further analyses. We then performed a k-means clustering analysis for the expressed core genes. We set the k value as 8 and 6 for analyzing stress and tissue related RNA-seq data, respectively. Nucleotide diversity as well as Fst were downloaded from WGVD [33].

To profile epigenetic modifications of core genes, we calculated the normalized read counts, representing the intensity of each mark, across 2 kb upstream and downstream of the expressed core gene promoters using deeptools followed by the k-means cluster analyses [34]. We used deeptools’ plotHeatmap module to plot the heatmap of chromatin features. Expression index was calculated by dividing the number of each expressed core gene (FPKM > 1) in a given tissue by the number of all tissues. The GO enrichment for each gene cluster was performed using agriGO v2.0 http://systemsbiology.cpolar.cn/agriGOv2/ (accessed on 1 May 2022) [35].

### 2.3. Co-Expression Analyses and Construction of Gene Regulatory Networks

We extracted 263 sets of RNA-seq expression matrices from the grass root database https://opendata.earlham.ac.uk/wheat/under_license/toronto/ (accessed on 1 May 2022) as TF_collobrative https://github.com/hwei0805/TF_CollaborativeNet (accessed on 1 May 2022) input files, and constructed a gene co-expression network with TF-centered in C7/C8. We then genetically selected a module from the co-expression network related to spikelet development. The R package GENIE3 (v1.12.0) [36] was used to infer GRN. The input expression matrix was the same as the one used in co-expression. Finally, core TFs, which are present in both co-expression networks as well as gene regulatory networks, can serve as core modules for regulating spikelet development. All networks were visualized using Cytoscape (v3.6.1) software [37]. Source codes can be found in our previous publication [38].

### 2.4. Statistical Analyses

All statistical analyses were performed using the basic statistical functions of the R computing environment. Unless otherwise stated, all statistical tests were performed using the Mann-Whitney U test, where * indicates that *p* < 0.05, ** means *p* < 0.01.

## 3. Results

### 3.1. Identification of the Core Genome of Common Wheat

To identify core genes in wheat, we analyzed a total of 16 sequenced genomes of wheat (Appendix A) by using the annotated protein sequences of Chinese Spring v1.1 as a reference [39]. We identified 62,299 core genes (Appendix A), accounting for ca. 54% of genes in Chinese Spring, using the parameters as below: genes with the similarity of protein sequences greater than 90%, length greater than 80%, and shared by at least 80% of the genome (Appendix A) [29]. After looking into subgenomes of common wheat, we found that there were 38%, 33% and 29% of core genes distributed in D, A and B subgenome, respectively (Figure 1A). To estimate genetic variations of these core genes among wheat population, we calculated the differentiation index (Fst values) of these genes between modern and local varieties and found that the majority of core genes had the Fst values of around 0.04 (Figure 1B), indicative of low genetic variations within population. After a closer examination, we detected 2945 core genes with Fst values greater than 0.25, representing the occurrence of large genetic differentiation within population [40]. To assess the biological relevance of these genetically variable core genes, we performed GO enrichment analyses and found that they were mainly enriched with GO terms associated with ADP/ATP/trans factor/7S RNA binding protein and DNA binding transcription factors (Figure 1C). We next compared gene length, gene expression levels, and nucleotide diversity between core and non-core genes. The core genes were found to be expressed at higher levels, had longer length and lower nucleotide diversity as compared to non-core genes (Figure 1D–F), in agreement with the results of *B. oleracea* [10] and *B.distachyon* [11]. In addition, most of the core genes had homolog genes in maize or *Arabidopsis*, indicating that they are functionally conserved in plants (Figure 1G) [11]. Transcription factors (TFs) play an important role in the regulation of gene transcription during plant growth and development [41]. We performed a core TF gene-related enrichment analysis and listed them from the right to left in a descending order of enrichment levels (Appendix A). We found that there were 25 TFs enriched within these core genes and *LFY* pioneer TFs [42] had the highest enrichment levels, indicating that they are the most conserved ones among population (Appendix A).

Collectively, through analyzing multiple genomes of wheat, we identified ca. 60,000 core genes with a low genetic differentiation level, high expression levels, long gene length, and low nucleotide diversity among the wheat population.

### 3.2. Core and Non-Core Genes Have Different Roles in Subgenome Differentiation

Common bread wheat is a result of a natural hybridization event between *Triticum dicoccoides* (AABB) and *Triticum tauschii* (DD) [43], therefore, the role of these core and non-core genes in subgenome differentiation remains to be investigated. We first compared the expression levels of core and non-core genes in each subgenome. As shown in Figure 1E, the expression levels of core genes within the subgenomes were always higher than the levels of non-core genes (Appendix A). Interestingly, the expression levels of the core genes in the D_subgenome were significantly lower than those in the A_ and B_subgenomes. In contrast, the expression levels of non-core genes in the D_subgenome were significantly higher than those in the A_ and B_subgenomes (Figure 2A,B). These results imply that core and non-core genes may have different roles in subgenome differentiation. Meanwhile, nucleotide diversity of all core and non-core genes showed B > A > D subgenomes (Appendix A), which is consistent with previous findings [25]. However, we also found that the nucleotide diversity of core genes in the B_subgenome was significantly higher than that of non-core genes (Appendix A).

The large number of homologous gene pairs in wheat provides a good model for studying core and non-core genes, especially the presence of 1:1:1 genes. To simplify the analyses, we divided the homologous gene pairs into two subtypes: both genes are core genes, and only either of them is a core gene (Appendix A). The expression levels, GO terms, and KaKs of these two subtypes of gene pairs were compared separately, and as expected, there was no difference for the homologous pairs with both of core genes (Figure 2C,E and Appendix A). However, between homologous gene pairs, the expression levels of core genes were significantly higher than those of non-core genes (Figure 2D). We also found that non-core genes had higher selection pressure (Figure 2F). Most importantly, although the majority of core and non-core genes functions were similar, we found some important functional differences between homologous genes. For example, core genes were exclusively enriched for cell cycle phase transition/DNA replication initiation (Figure 2G); whereas non-core genes were enriched exclusively for the mitochondrial cell cycle (Figure 2H). We also compared the GRO-seq levels for different subtypes of genes, and in general, higher gene expression levels were associated with higher GRO-seq signals (Figure 2I–L), consistent with previous results [44].

Taken together, our results demonstrate that core and non-core genes have different roles in subgenome differentiation.

### 3.3. Potential Biological Relevance of Core Genes in Wheat

To interrogate the potential biological implications of these core genes in wheat, we profiled their expression levels in multiple tissues or various stresses using public gene expression data [32]. We obtained 34,134 expressed genes and ~25,000 non-expressed ones. We then conducted a k-means clustering assay for these expressed genes and obtained eight sub-clusters related to various treatments (Figure 3A) and six sub-clusters related to the tissue development examined (Figure 3B). For example, we found that 5103 genes in C5 and 5626 genes in C6 in Figure 3A were highly expressed under heat and salt treatment, respectively (Figure 3A); 8898 genes in C1 and 9477 genes in C6 in Figure 3B were highly expressed in the different developmental stages of spikelets (Figure 3B). It has been documented that wheat is sensitive to heat and salt stress that can cause a dramatic reduction in wheat yield [45,46]. We then specifically focused on core genes exhibiting inducible expression under heat or salt conditions. As shown in GO term enrichment results, GO terms related to photosynthesis were significantly enriched under the salinity (Figure 3C) and heat (Figure 3D) conditions, which is consistent with the previous finding that both stresses affect photosynthesis in wheat [47,48]. Moreover, we also found that GO terms related to responses to calcium ions/metalion/osmotic stress and cellular ion homeostasis were enriched under heat stress (Figure 3D). We also conducted similar GO term enrichment analyses for genes related to tissue development, especially for genes highly expressed in spikelets that are closely related to yield potentials [49]. We found that genes highly expressed in spikelet_I (Figure 3E) and spikelet_II (Figure 3F) were overrepresented in GO terms related to the transition from asexual to reproductive stages of meristematic tissues, meiosis, and plant epidermal cell differentiation, which are essential for tissue development in sexual plants [50,51]. Impacts of high temperature on spikelet development have been extensively studied in rice, maize, and wheat [52,53,54]. It is still unclear how the core genes function in both heat stress and spikelet development. To address this, we identified 3376 genes highly expressed in spikelets under heat stress, including *HSP26*, *TaWRKY33 TaGRF-2D* genes already reported in relation to the heat responses [55,56], and the starch synthesis genes *SSII-1* and DSB repair proteins *TaRPA1a,* indicating potential effects of high temperature on yield and quality of wheat (Figure 3G, Appendix A). Again, we conducted a GO term enrichment assay and found that they had enriched GO term functions related to biological processes such as RNA processes, chromosome organization, and the regulation of gene expression. More importantly, we also observed GO terms responsible for some critical biological processes such as photo response/morphogenesis, female gamete production, embryo sac oocyte differentiation, and telomere maintenance (Figure 3H), suggesting the important roles of these core genes in the heat responses and spikelet development in wheat.

Taken together, all these results show that a subset of core genes may play important roles in stress responses and/or tissue development in wheat.

### 3.4. Core Transcription Factors Enriched with H3K27me3 and H3K4me3

Histone modifications play fundamental roles in various biological processes through the modulation of gene transcription [57]. The chromatin features of core genes are still less studied in plants. To answer this question, we profiled the read density of DNase-seq (DHSs) and six histone marks derived from wheat seedlings [58] across ± 2 kb of the TSS of 34,134 expressed genes followed by k-means clustering analyses (Figure 4A). We then divided them into eight sub-clusters (C1–C8) in terms of distinct profiling of histone marks and DHSs (Figure 4A). We found that C1, C7 and C8 had a relatively high percentage of core TFs, accounting for 19%, 24% and 27% of the total core TFs, respectively (Figure 4B). After a closer examination, we found that genes in C7 and C8 tended to be co-presence of both the repressive mark H3K27me3 and the active mark H3K4me3, the proportion of which was significantly higher than that of genes with similar chromatin features in the whole genome (Figure 4C and Appendix A), indicative of bivalency of these genes. After calculating the expression levels (FPKM) of genes in each cluster, we found that genes in C7 and C8 had lower expression levels than genes in other clusters most likely due to the presence of relatively high level of repressive mark, H3K27me3 (Figure 4D). We also found that approximately 93% of these non-expressed core genes were enriched with H3K27me3 marks (Appendix A). It has been documented that genes with a bivalent H3K4/27me3 mark play vital roles in tissue development and stress responses in plants [59,60]. To assess possible roles of C7 and C8 related core genes in the tissue specific regulation, we calculated the gene expression index in each subcluster and found that the expression breadth of core genes in C7 and C8 was significantly lower than that of core genes in other subclusters. This indicated that core genes in C7 and C8 had higher tissue specificity than other core genes (Figure 4E,F). We then performed GO enrichment analyses for core genes in C7 and C8 and found that both categories were indeed associated with GO terms related to stimuli responses, and growth and development (Appendix A). 

Collectively, all these results show that expressed core genes exhibit distinct chromatin features. In particular, ca. 50% of TFs are enriched with H3K4/27me3 bivalent mark and possibly function in the tissue specificity in wheat.

### 3.5. Construction of Core TF-Centered Regulatory Networks 

TFs play vital roles in the regulation of gene transcription [61,62]. To investigate how core TFs function in the regulation of tissue development or stress responses in wheat, we conducted co-expression assay and obtained six co-expression modules containing the majority of genes in C7 and C8 mentioned in Figure 4B using the procedures described before [63] (Figure 5A). To identify modules associated with the tissue development, we performed a k-means clustering analysis for checking the expression levels of core TFs in different tissues using public RNA-seq data [64] (Appendix A). Spikelet development is an important agronomic trait that is closely related to grain yield and quality [49]. To visualize the involvement of core TFs in the regulation of spikelet development, we specifically extracted all genes highly expressed in spikelets and found a co-expressed module containing most of these genes. It mainly contains *GRAS*/*MYB*/*bZIP*/*ERF*/*Trixhelix* TFs (Figure 5B, Appendix A). We found that the gene *TraesCS3A02G233000/TraesCS3B01G262400* is homologous to the GRAS family *OsSLRL1* gene (Figure 5B), which encodes a DELLA protein and was highly expressed in rice germ cells and elongated stalks but lowly expressed in rice leaves and roots; its loss of function mutant exhibits defects in rice flowers [65,66]. The gene *TraesCS5A02G314600/TraesCS5B02G315500* is homologous to the *OsEATB* gene that can promote rice tillering and panicle branching [67]. In addition, it was reported that *TraesCS3A02G233000*/*TraesCS3B01G262400* was expressed in wheat leaves under salt stress [68], implying that this TF may be involved in spikelet development and responses to the salt stress through co-expression network. To examine the regulatory relationship between core TFs, we further constructed a core TF centered regulatory network by using GENIE3 [36] and selected the top 1000 predicted regulatory relationships for visualization (Appendix A). We found that it contains 9 of the 24 core TFs in the co-expression module, including the hub TF *TraesCS3D02G118300* (Figure 5C). We also found that a salt-inducible TF *TraesCS7B02G075600* [68] is located at the upstream of *TraesCS3D02G118300,* the promoter region of which indeed contains a binding motif (GCCACGTG) of *TraesCS7B02G075600*. This result indicated that *TraesCS7B02G075600* may regulate the expression of *TraesCS3A02G233000/TraesCS3B01G262400* through *TraesCS3D02G118300*. (Figure 5C). 

In summary, these results indicate that a subset of core TFs are involved in spikelet development and stress response through a TF-centered regulatory network.

## 4. Discussion

The core genome of a species contains core genes present in the vast majority of subspecies and common regulatory genomic DNA elements, thereby involving in some essential biological processes in mammals and plants [17,69]. For example, it has been reported to maintain the fundamental cellular events of an organism, including DNA replication, translation and maintenance of cellular homeostasis [3,11]. However, intrinsic features and biological implications of wheat core genes are still less studied. In this study, we provided evidence showing that cores genes had some distinct features as compared to non-core genes, including longer gene length, more conserved among populations, higher expression levels and lower selection pressure (Figure 1D–F). Similar findings have been reported in other plant species such as *B. distachyon*, maize and soybean [8,9,11], indicating conservation of these intrinsic features among plant species. Our study showed that core genes accounted for ca. 54% across 16 wheat varieties examined, which is 10% less than the previous study (ca. 64%) [24]. This difference is possibly caused by using different reference genome and the different methods for identification of core genes between two studies. When compared with findings reported in other species, we observed variations in the percentage of core genes, including ca. 50% in soybean [9], ca. 49% in wild soybean [70], ca. 48% in rice [6], ca. 58% in maize [8] and ca. 55% in *B. distachyon* [11], ca. 44% in strawberry [14], ca. 62% in *B. napus* and cotton [12,13], ca. 81% in *B. oleracea* [10], ca. 74% tomato [15] and ca. 75% in sunflower [71]. This variation could be a species-dependent or caused by the number of accessions used for analyses. Moreover, our study showed that most of core genes were conserved among different species, including *Arabidopsis* and maize (Figure 1G), indicating that their functions are conserved and essential for the growth and development of all plant species [17]. In addition to these common functions, we found that selection pressure and nucleotide diversity were not identical between core and non-core genes even among homologous gene pairs, implying that core as well as non-core genes may have different roles in subgenome differentiation (Figure 2). We also found that the core genes might play vital roles in the regulation of tissue development and stress responses (Figure 3). Similarly, the involvement of core genes in immune system and reproductive stage was reported in soybean [9]. These results suggest that the core genes are indispensable for ensuring normal growth and development as well as stress responses in the plant kingdom. 

In addition to the direct involvement of core genes in some biological processes, our study for the first time provided evidence showing that some of core TFs can function in the regulation of gene expression through the hub TF-centered regulatory network, indicative of their indirect involvement in some biological processes, especially for some key agronomic traits. The C7/C8 contains 24 core TFs (Figure 5), some of which were found to be involved in the regulation of spikelet development and stress response. This indicates that indirect functions of some core TFs can be mediated by the regulatory network. A number of genes exhibited high DNA sequence similarity with already reported genes in rice, which are involved in the regulation of both spikelet development and stem growth [65,66,67], including *OsSLRL1* homolog gene *TraesCS3A02G233000/TraesCS3B01G262400* and *OsEATB* homolog *TraesCS5B02G315500/TraesCS5A02G314600* (Figure 5B,C). *KRN2* and *OsKRN2* genes in rice and maize have recently been reported to enhance the yield of both crops through editing the promoter region [72]. A ~4-kb substitution in the promoter region of core TF *TomLoxC* resulted in a change of the tomato flavor [15]. Thus, by integrating the resequencing results or gene editing, this study can also provide some key core TF candidates for crop breeding towards favorable agronomic traits. 

Histone modifications play an important role in the modulation of gene expression, thereby involving in various biological processes in plants [73], including the regulation of flowering time [74], biotic and abiotic responses [75,76,77]. However, chromatin features of core genes are still understudies in plants. Here, we found that wheat core genes can be divided into eight sub-clusters in terms of distinct chromatin states, a combination of DHSs and six histone marks (H3K4me3, H3K27me3, H3K36me3, H3K4me1, H3K9me2 and H3K9ac). Moreover, we found that genes in C7 and C8 were mainly associated with H3K4me3 and H3K27me3 (Figure 4), indicative of their bivalency feature. More importantly, we found that genes of both sub-clusters tended to have tissue-specific functions and be stress responses. Genes with enrichment of H3K4me3 and H3K27me3 covalent marks have been reported to be involved in stress responses and stress memory in plants, such as the cold response in potato [59], drought stress and stress memory in *Arabidopsis* [78], and sporophyte and gametophyte development in *Arabidopsis thaliana* [79]. These results indicate that functional differentiation of core genes can be mediated by chromatin features.

## 5. Conclusions

In conclusion, our study for the first time characterizes expression and epigenetic features of core genes in common wheat, thereby providing a valuable resource for genetic and epigenetic studies of core genes in wheat.

## Figures and Tables

**Figure 1 genes-13-01112-f001:**
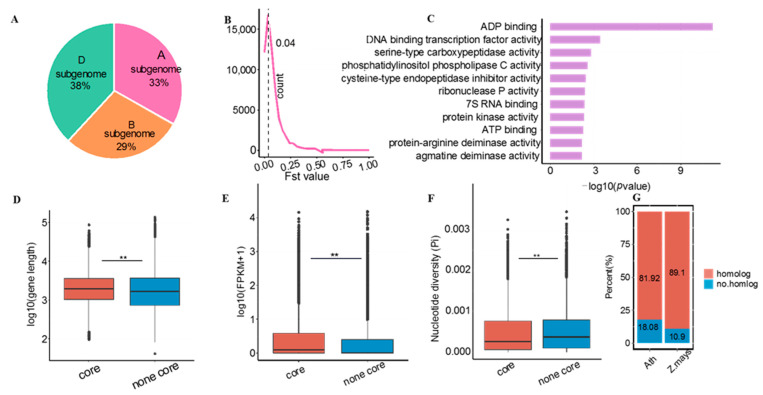
Identification of core genes in wheat. (**A**), The percentage of core genes distributed in subgenomes, 33% for A_subgenome, 29% for B_subgenome and 38% for D_subgenome. (**B**), Population differentiation index of core genes between cultivars and landraces, dashed line represents Fst = 0.04. (**C**), GO enrichment analyses of core genes with a population differentiation index greater than 0.25. (**D**–**F**), Comparisons of core and non-core gene lengths (**D**), expression levels (**E**), and nucleotide diversity (**F**). (**G**), Proportion of core genes with homolog genes in maize and *Arabidopsis*. Where ** means *p* < 0.01.

**Figure 2 genes-13-01112-f002:**
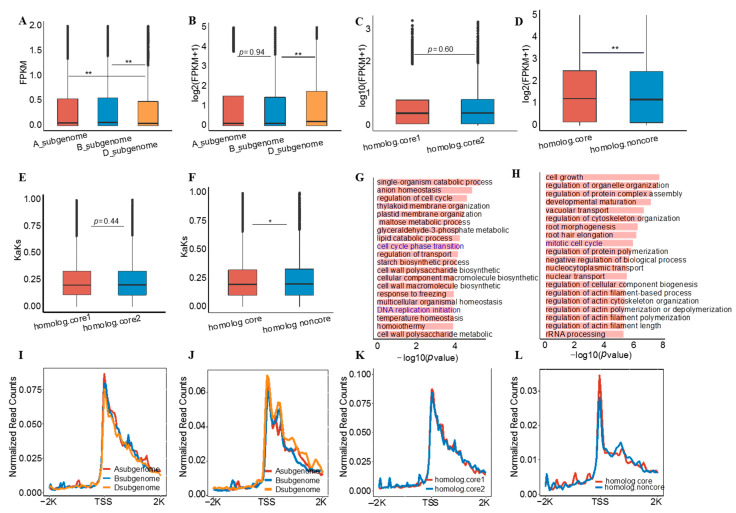
Core and non-core genes have different roles in subgenome differentiation. (**A**,**B**), (**A**) Comparing the expression levels of all core genes and (**B**) all non-core genes within subgenomes. (**C**–**F**), Comparisons of expression levels and KaKs values between paired homologous genes. where (**C**,**E**) indicate that both genes are core genes, and (**D**,**F**) indicate that one gene is a core gene, and another is non-core genes. (**G**,**H**), GO enrichment analyses of core and non-core genes in homologous gene pairs, (**G**) indicates GO terms specific to the core genes; (**H**) indicates GO terms specific to the non-core genes. Only the top 20 most important GO terms are listed. (**I**–**L**), GRO-seq read the intensity of all core genes (**I**), all non-core genes (**J**), all homologous pairs as core genes (**K**), and only one homologous pair as core gene (**L**). In general, GRO-seq signals and gene expression levels were positively correlated. Where * indicates that *p* < 0.05, ** means *p* < 0.01.

**Figure 3 genes-13-01112-f003:**
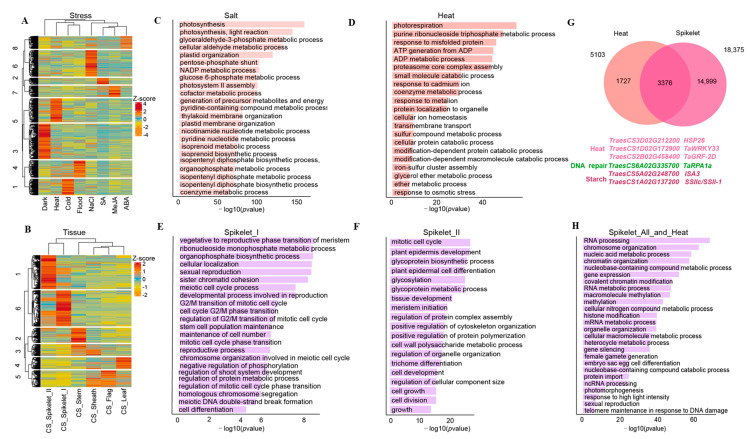
Expression patterns of core genes in abiotic stresses and multiple tissues. (**A**), The expression level clustering result of core genes in the stress treatment. (**B**), The expression level clustering result of core genes in different tissues, all clustering results were transformed by Z-score, and the genes with the sum of FPKM less than three were excluded for further analyses. (**C**–**F**), GO enrichment analyses of different genes. GO enrichment analyses of heat stress (**C**), salt stress (**D**) and core genes with high expression levels in spikelet_I (**E**) and spikelet_II (**F**). (**G**), Venn diagram of genes highly expressed in heat stress and genes highly expressed during the spikelet development, genes listed below the Venn diagram indicating reported genes related to heat response, starch synthesis and DNA repair. (**H**), GO enrichment analyses of highly expressed genes in heat stress and spikelet development.

**Figure 4 genes-13-01112-f004:**
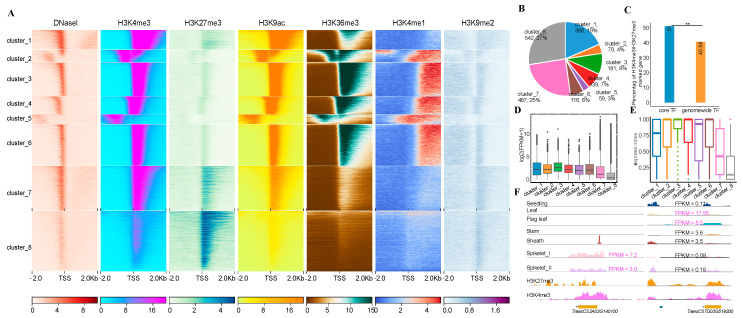
Epigenetic landscapes of expressed core genes. (**A**), Chromatin states were performed on the upstream and downstream 2 kb of the expressed core gene promoters. A total of eight subcategories were classified, C7/C8 was enriched with H3K27/K4me3. (**B**), The proportion of TFs in different chromatin states, C7/C8 occupied 50% of the TFs. (**C**), Comparisons of core TFs with H3K4/K27me3 bivalent mark between C7/C8 and the whole genome. (**D**,**E**), Analyses of the core gene expression values (**D**), and the expression breadth (**E**) corresponding to different chromatin states. Genes with higher active histone modifications correspond to higher gene expression levels and lower tissue specificity, and vice versa. (**F**), IGV illustrates genes for spikelet-(**left**) and leaf-(**right**) specific expression, respectively. Where ** means *p* < 0.01.

**Figure 5 genes-13-01112-f005:**
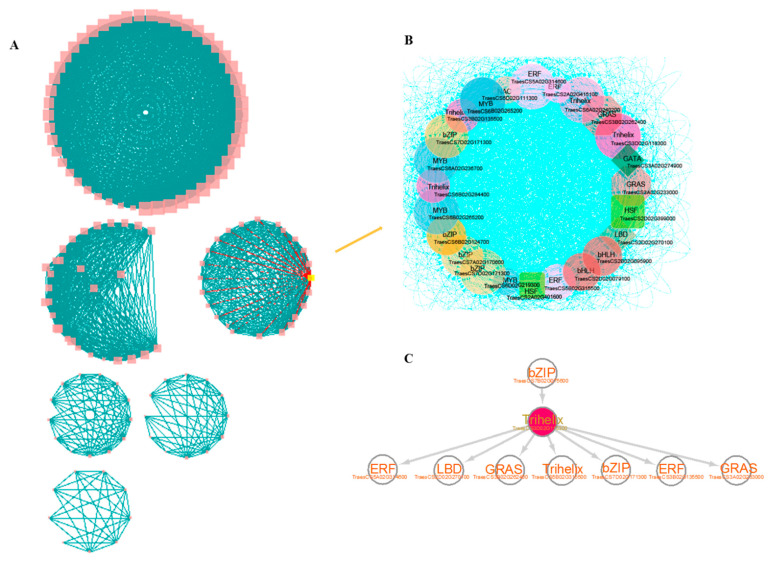
Construction of TF-centered regulatory networks. (**A**), Construction of TF related co-expression network using H3K4/K27me3 marked TFs. (**B**), Modules related to spikelet development were selected from the co-expression network. (**C**), A regulatory network with the regulation direction was re-constructed using GENIE3 for the TFs in the selected modules, and the top 1000 weighted regulatory relationships were selected, 9 TFs of which were present in the co-expression modules.

## Data Availability

All data used in this study were obtained from public data. the Gene Expression Omnibus Database accession numbers are as follows: GRO-seq:GSE178276 [44]. ChIP-seq and DNase-seq data and tissue development, and stress-treated RNAseq data of bread wheat were published previously under accession GSE139019 and GSE121903 [32,58]. 263 sets of RNA-seq expression matrices from the grass root database: https://opendata.earlham.ac.uk/wheat/under_license/toronto/ (accessed on 1 May 2022).

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
