# Peer review of "Characterization of Expression and Epigenetic Features of Core Genes in Common Wheat"

_genes, 2022, doi:10.3390/genes13071112_

Round 1

Reviewer 1 Report

This paper is a bioinformatics analysis of data deposited in gene banks. It includes some interesting ideas on the use of available core and non-core gene data, their analysis and attempts to explain the results obtained. It is good that the authors have undertaken such analyses in wheat, because compared to rice or barley (where such studies are more advanced) this is an almost unexplored area.

1. I find the results describing the different roles of core and non-core genes in genome differentiation interesting.

2. An interesting result concerns the demonstration that expressed core genes show distinct chromatin features. Here, however, additional work is required.

However, I have one important question. Based on what criteria did the authors take a threshold for selecting genes with FPKM <3? The use of this threshold determines the number of genes that will be analyzed.

Author Response

This paper is a bioinformatics analysis of data deposited in gene banks. It includes some interesting ideas on the use of available core and non-core gene data, their analysis and attempts to explain the results obtained. It is good that the authors have undertaken such analyses in wheat, because compared to rice or barley (where such studies are more advanced) this is an almost unexplored area.

  1. I find the results describing the different roles of core and non-core genes in genome differentiation interesting.
  2. An interesting result concerns the demonstration that expressed core genes show distinct chromatin features. Here, however, additional work is required.

Response: We highly appreciate your positive feedback. Thank you!

However, I have one important question. Based on what criteria did the authors take a threshold for selecting genes with FPKM <3?

Response: We thank the reviewer for pointing this out. We followed the standard described in the published literature (Wang et al., Plant Cell, 2021, 33: 865-881), in which genes with FPKM > 3 were set as expressed ones in wheat. We used the same datasets (RNA-seq and ChIP-seq) for analyses. To get more expressed genes in tissues or treatments, we set the sum of expression values (FPKM) as 3 for identification of expressed genes.

The use of this threshold determines the number of genes that will be analyzed.

Response: Following your suggestions, we have re-analyzed the epgenetics landscapes of these non-expressed genes, we found that about 93% of the genes were enriched with H3K27me3. We added the new analyses in Figure S4B, 4D. We also added some descriptions in the text.

Reviewer 2 Report

Genome sequencing and its analyses is a very labor - intensive task. But over the last decades a lot of data has been accumulated in available databases. Normally researchers analyze and discuss their own results. That’s why the presented article is of special interest comparing genome sequences of 16 different wheat varieties. All together more than 62 thousands core genes were identified and classified into 8 subclusters with different epigenomic features. Also, according to their expression profiles, analyzed core genes were classified into genes related to tissue development and stress responses. Furthermore, in comparison with other genes, they are bigger and more conserved among populations; have higher expression levels and lower selection pressure, which theoretically could be predicted.

Generally the presented article is logically structured, written in good English and has enough illastrations.

Author Response

Genome sequencing and its analyses is a very labor - intensive task. But over the last decades a lot of data has been accumulated in available databases. Normally researchers analyze and discuss their own results. That’s why the presented article is of special interest comparing genome sequences of 16 different wheat varieties. All together more than 62 thousands core genes were identified and classified into 8 subclusters with different epigenomic features. Also, according to their expression profiles, analyzed core genes were classified into genes related to tissue development and stress responses. Furthermore, in comparison with other genes, they are bigger and more conserved among populations; have higher expression levels and lower selection pressure, which theoretically could be predicted.

Generally the presented article is logically structured, written in good English and has enough illastrations.

Response: We greatly appreciate your affirmation on our work. Thank you!

Reviewer 3 Report

The paper deals with the analysis of genome sequences from 16 different wheat varieties and identified core and non-core genes with different roles in sub-genome differentiation.   As mentioned by the authors, the study provided a valuable resource for functional characterization of core genes in stress responses and tissue development in wheat.

The paper is well written and provides interesting results which need minor revision as shown in the attached file.

Author Response

The paper deals with the analysis of genome sequences from 16 different wheat varieties and identified core and non-core genes with different roles in sub-genome differentiation.   As mentioned by the authors, the study provided a valuable resource for functional characterization of core genes in stress responses and tissue development in wheat.

The paper is well written and provides interesting results which need minor revision as shown in the attached file.

Response: We greatly appreciate your affirmation on our work. We made all changes following your suggestions. Thank you!

I suggest to split this Figure 1 to 3-4 more related items. It is difficult to follow in its current presentation.

Response: We thank the reviewer for pointing this out. We totally agree with you. Following the suggestion, we moved original Figure 1A and 1I to Figure S1 and S2, respectively.

Please make 1A easily understandable

Response: Following the suggestion, we simplified the flow chart for the identification of core genes in wheat and changed it as Figure S1.

Could you please give any criteria or basis why and how these 16 varieties were selected?

Response: Thank you for your question. In previously, pan-genome work in wheat has largely relied on the unassembled genome. Here, we mainly use these chromosome-level genomes for core gene annotation. Our criterion therefore relies on current genome availability.

the use of A, B and D as figures sub-label might cause confusion to readers with A, B and D genomes

Response: We thank the reviewer for pointing this out. We agree with you. To avoid confusion between A/B/D subgenome with A/B/D as figure sublabel, we changed A/B/D subgenome as A_/B_/D_subgenome.

If the % of Brassica and cotton are the same, then combine both references {13 and 14}

Response: We combined both references as suggested. Thank you!

I think if you can give a short conclusion statement it will be better 

Response: We added a short conclusion statement as suggested. Thank you!